# Magnitude of job satisfaction and intention to leave their present job among nurses in selected federal hospitals in Addis Ababa, Ethiopia

**Aynye Negesse Woldekiros** [1]*, **Elsabet Getye**[2], **Ziyad Ahmed Abdo** [3]*

1 Department of Public Relation and Communication, Ethiopian Ministry of Health, Addis Ababa, Ethiopia,
2 Department of Public Health, St. Paul's Hospital Millennium Medical College, Addis Ababa, Ethiopia,
3 Department of Hygiene and Environmental Health, Ethiopian Ministry of Health, Addis Ababa, Ethiopia

* aynyenegesse@gmail.com (ANW); ziyadahm1982@gmail.com (ZAA)

**Data Availability Statement:** Data are available at the protocols.io repository at: dx.doi.org/10.17504/protocols.io.b5b6q2re (https://protocols.io/view/

## Abstract

### Background

Job dissatisfaction issues and health workers' intention to leave is an increasing problem that threatens the function of the health care sector worldwide, especially in developing countries, including Ethiopia. As part of future evidence, this study was intended to assess nurses' job satisfaction and intention to leave their current work and its associated factors in federal public hospitals in Addis Ababa, Ethiopia.

### Method

An institutional based cross-sectional study design was used to conduct the study. A simple random sampling technique was employed to select 408 nurses from selected federal hospitals in Ethiopia. Self-administered questionnaire was used to collect the data. Logistic regression was employed to identify the predictor variables with consideration of statistical significance at P <0.05 adjusted odds ratios calculated at 95% CIs.

### Result

The magnitudes of job satisfaction and intention to leave their current job among nurses in this study were 47.7% and 80.6%, respectively. Salary is imbalanced with demands [AOR = 2.85 (1.24, 6.57)], managers who have no personal plan for developing skills [AOR = 3.74 (1.58, 8.87)], stressful jobs [AOR = 0.28 (0.11, 0.71)], health problems are a reason for having thoughts about changing jobs [AOR = 3.02 (1.17, 7.79)], and a lack of development [AOR = 4.13 (1.51, 11.3)] were identified as determinant factors for intention to leave.

### Conclusion

The overall intention to leave their current job among nurses working in selected federal hospitals in Ethiopia was high. The government of Ethiopia should balance the salary of nurses

magnitude-of-job-satisfaction-and-intention-to-lea-b5b6q2re.html).

**Funding:** The author(s) received no specific funding for this work.

**Competing interests:** The authors have declared that no competing interests exist.

with the current market level. Hospital leaders should plan the way nurses develop their educational and job carrier levels.

## Introduction

Intention to leave is defined as an employee's plan to quit the present job and look forward to finding another job in the near future [1–3]. The widespread nursing shortage and nurses' high turnover rate have become the global issues [4] and have an adverse impact on health systems around the world [2, 5]. Losing a strong employee who has invested time and money in recruiting and training can cost a lot [2]. Job satisfaction is the most consistent predictor of nurses' intention to leave and has been reported to explain most of the variance in intention to leave [6–8]. Employee job satisfaction is the fulfillment, gratification, and enjoyment that come from work. It is not just the money or the fringe benefits but the feelings employees receive from the work itself [9].

Globally, the shortage of nurses is a challenging issue among the health sectors [3, 10]. The negative consequence of high turnover includes costs associated with recruitment and orientation of new health professionals, loss of experienced one, and potential for increasing an adverse patient outcome and reduced organizational performance [11–13], which impacts the capacity to meet patient needs and provide quality care [9].

Health worker turnover is an increasing problem that threatens the function of the health care sector worldwide, especially in developing countries [11]. Asian countries show a high level of intention to leave their current jobs. A study performed in China Shangie showed that 50.2% of nurses were dissatisfied, and 40% of nurses had an intention to leave their current employment [4]. A study performed in Riyad, Saudi Arabia, showed that almost 94% indicated turnover intention from their current hospital [14]. A similar study performed in Iran showed that 69.3% of nurses were dissatisfied with their work life [15].

The magnitude of intention to leave across African countries ranged from 18.8 to 41.4% [5, 16], and in Ethiopia, it ranged between 50 and 83.7% [1, 12, 17–20]. While Sub-Saharan Africa (SSA) is home to 12% of the global population, it only employs 3.5% of the global health workforce, which is disproportionate to 27% of the global burden of disease on the continent [21]. Although the WHO recommends 2.5 health care workers per 1000 people [21], in Ethiopia, there are only 0.84 per 1,000 people [22].

Although factors affecting intention to leave one's current job among health professionals vary over time, many studies have identified age, sex, marital status, level of education, satisfaction with autonomy and professional opportunities, scheduling, pay and benefits, workload, working environment, opportunity for professional growth, development staff relationship, supportive supervision, high continuance commitment and high normative commitment as significant predictors of intent to stay in the nursing profession standard of care and workload [2, 6, 8, 9, 14, 18, 23]. In general, work satisfaction comprises intrinsic and extrinsic factors [9]. Intrinsic factors are internally derived and include personal achievement, sense of accomplishment, and prestige [20, 23]. Extrinsic factors are those derived from factors related to pay and benefits, working conditions, and resources [9, 15, 24].

However, the problem of intention to leave the current job among health care professionals is high worldwide, especially in developing countries, including our country, and there are limited and updated studies done in Ethiopia. To reduce the problem, individual and group efforts should be supported with evidence. As such, continuous studies should be performed

to generate strong evidence that supports tackling the problem. As part of future evidence, this study intends to assess nurses' job satisfaction and intention to leave their work and the associated factors in federal public hospitals in Addis Ababa, Ethiopia.

## Method and material

### Study design and settings

An institutional-based cross-sectional study design was used to conduct the study among nurses at St. Paul's and St. Peter's hospitals. They built and started giving services in 1977 and 1971, respectively. According to their official page, St. Paul's Hospital and Millennium Medical College currently has more than 2,500 clinical, academic and administrative staff members. Among the total nurses in the hospital, 170 were diploma nurses, and 511 were BSc nurses. The current St. Peter's Hospital was established in 1971 as a TB treatment center. The hospital has a total of 500 staff members, of which 250 are health care professionals. The study was conducted from October 21, 2020, to November 10, 2020.

### Population and eligibility criteria

All nurses working in federal hospitals in Addis Ababa, Ethiopia, were considered the source population. Nurses working at St. Paul's Hospital Millennium Medical College and St. Peter specialized hospitals were considered the study population. All nurses with a qualification level of diploma or above who were currently working in the hospitals were included in the study. Those who were seriously ill during the data collection period were excluded from the study.

### Sample size and sampling strategy

The required sample size for the study was calculated using a single population proportion formula with assumptions of 95% confidence level, 5% margin of error and a proportion of 59.4% from a study done in Ethiopia to explore the intention to turnover of nurses working in governmental health care institutions in East Gojjam [25]. Therefore, based on this information, the calculated sample size was 371. By considering a 10% nonresponse rate, the final sample size for this study was considered to be 408.

From each selected federal hospital, the number of nurses was obtained from monthly payrolls. Then, based on the number of nurses in each hospital, the sample size was proportionally allocated to the hospitals. Finally, a simple random sampling method was carried out to select the study participants from all categories of nurses in each hospital.

### Study variables

- **Dependent Variable:** Intention to leave

- **Independent Variables:** Socio-demographic characteristics such as age, sex, marital status, education status, years of experience at work, role or position in the health care facility and primary work area in the hospital were included. Organizational career-related characteristics such as job satisfaction, cause of turnover intention, working experience, compensation, salary, working hours, support from management, communication and personal plan were also included. Work environment-related characteristics such as payment, staffing, nursing education, stressful job, schedule/working hours, career, health, development, support from coworkers, organizational policies, work condition, and supervision were also included in these independent variables.

### Data collection tools and quality control

A self-administered questionnaire was used to collect quantitative data. The tool was developed after reviewing similar studies and modified in line with the objective of this study. Prior to actual data collection, the questionnaire was adjusted and corrected based on the pretest result, and the final questionnaire was translated to Amharic and then back to English to ensure consistency. In the process of data collection, six diploma holder data collectors and two degree holder supervisors were used. The completeness of the collected data was checked by the supervisor and principal investigator on a daily basis. The checking system of data was on a daily basis for completeness.

### Data management and analysis

The data were entered into Epi-Data version 3.1 and exported to SPSS version 22 for data management and analysis. The general characteristics of the study population and the magnitude of intention to leave were described by using descriptive summary statistics such as frequencies, standard deviations, means, percentages and texts. Hence, logistic regression was employed to identify the predictor variables. Statistical significance was considered at $P < 0.05$ with adjusted odds ratios calculated at 95% CIs.

### Ethics approval and consent to participate

Before conducting the study, ethical clearance was obtained from St. Paul's Hospital and St. Peter's Hospital ethical review board. Participants' right to self-determination and autonomy was respected, and study participants were given any information they needed verbally and in written prior to being involved in the study. The rights of each respondent refusing to answer a few or all questions were respected. Participation was voluntary, and participants could withdraw from the study at any time without explanation.

## Result

### Socio-demographic characteristics of respondents

A total of 392 nurses participated in the study, for a response rate of 96.1%. The majority (69.1%) of the respondents were females. The mean age of the respondents was 29.8±4.7 years. Nearly half (49.2%) of the participants were married. More than three-fourths (81.6%) of the participants were BSc nurses. Nearly half (45.2%) of the participants had three to five years of work experience. Six out of ten participants were staff nurses in their role, and 72 (18.4%) of the participants' primary working areas in the study area were operating rooms (Table 1).

### Organizational and career related characteristics

Nearly nine out of ten (88.5%) participants rated that the cause of turnover intention of their organization was high. Approximately 319 (81.4%) nurses agreed that their work experience possessed their personal skills and helped them stay in their organization. More than half (59.7%) of the study participants responded that being dissatisfied with the amount of compensation paid by their organization was one of the reasons for which they intended to leave their current job. Approximately 239 (61%) of the participants raised that, as it interferes with social life, long working hours were also another cause for their intention to leave. Lack of support from managers (67.3%) and unclear interpersonal communication with their managers (57.1%) were also among reasons for their intention to leave the current job (Table 2).

**Table 1. Socio-demographic characteristics of nurses working in selected federal hospitals in Ethiopia, 2020 (N = 392).**

| Variables | Category | Frequency (n = 392) | Percent |
|---|---|---|---|
| Sex | Male | 121 | 30.9 |
| | Female | 271 | 69.1 |
| Age | 20–29 | 215 | 54.8 |
| | 30–39 | 161 | 41.1 |
| | $\geq 40$ | 16 | 4.1 |
| Marital status | Single | 148 | 37.8 |
| | Married | 193 | 49.2 |
| | Divorced | 29 | 7.4 |
| | Widowed | 22 | 5.6 |
| Educational status | Diploma | 12 | 3.1 |
| | Bachelor degree | 320 | 81.6 |
| | Master degree | 60 | 15.3 |
| Work experience | $\leq 2$ years | 35 | 8.9 |
| | 3–5 years | 177 | 45.2 |
| | $\geq 6$ years | 179 | 45.7 |
| Role or position | Staff nurse | 241 | 61.5 |
| | Head nurse | 121 | 30.9 |
| | Other | 30 | 7.7 |
| Primary work area | Medical ward | 55 | 14.0 |
| | Surgical ward | 45 | 11.5 |
| | ICU | 63 | 16.1 |
| | Emergency | 58 | 14.8 |
| | OR | 72 | 18.4 |
| | Other | 99 | 25.3 |

### Work environment-related characteristics

Regarding work environment-related factors, the majority (60.5%) of study participants were found their job stressful; however, most (83.2%) of them replied that they had support from coworkers. More than two-thirds and one-half of the study participants had thoughts about leaving an organization and leaving the profession, respectively. In addition, more than half (51.0%) and approximately two-thirds (65.3%) of study participants applied for a new job during the last year, and schedule/working hours were reasons why they had thoughts about quitting their current job, respectively. Approximately 201 (51.3%), 301 (76.8%), 241 (61.5%), 203 (51.8%), 246 (62.8%) and 283 (72.2%) respondents reason for having thoughts about quitting and changing the job were manager, organizational factors, career, health, lack of development and salary, respectively (Table 3).

### Magnitude of job satisfaction and intention to leave

Approximately 47.7% of respondents were satisfied with their current job. Among the 392 study participants, 316 (80.6%) had the intention to leave their current health institution (Fig 1).

### Factors associated with intention to leave

Bivariate and multivariable analyses were conducted to determine the association of one independent variable with the dependent variables. Variables with $P < 0.25$ during the bivariate

**Table 2. Organizational and career-related characteristics among nurses working in selected federal hospitals in Ethiopia, 2020 (N = 392).**

| Variables | Category | Frequency | Percent |
|---|---|---|---|
| The cause of turnover intention of the organization | High | 346 | 88.5 |
| | Low | 45 | 11.5 |
| The working experience did possess personal skill that helps to stay in the Organization | High | 319 | 81.4 |
| | Low | 73 | 18.6 |
| The satisfaction level with the amount of compensation paid by the current organization | Satisfied | 158 | 40.3 |
| | Dissatisfied | 234 | 59.7 |
| Male workers leave organization more than female workers | Yes | 208 | 53.1 |
| | No | 184 | 46.9 |
| Feel that chosen the wrong career | Yes | 226 | 57.7 |
| | No | 166 | 42.3 |
| Actively looking for jobs that will help to better career | Yes | 64 | 16.3 |
| | No | 328 | 83.7 |
| Salary is in balance to the demands employer places as an employee | Yes | 219 | 55.9 |
| | No | 173 | 44.1 |
| Working hours are hindrance for social life | Yes | 153 | 39.0 |
| | No | 239 | 61.0 |
| Experience lack of support from management | Yes | 128 | 32.7 |
| | No | 264 | 67.3 |
| Find support from the manager | Yes | 150 | 38.3 |
| | No | 242 | 61.7 |
| The communication from the manager is clear (instructions, tasks, goals and feedback) | Yes | 224 | 57.1 |
| | No | 168 | 42.9 |
| Feel that manager has a personal plan for developing skills/competence/qualifications | Yes | 207 | 52.8 |
| | No | 185 | 47.2 |
| When negative incidents occur, experience that the managers feedback strengthens for the future | Yes | 154 | 39.3 |
| | No | 238 | 60.7 |

analysis were included in the multivariate logistic regression analysis to determine the association of independent variables with intention to leave the current job by controlling for confounding variables. After computing multivariate analysis, managers had personal plans for developing skills and stressful jobs. Salary is the reason for changing jobs, health is the reason for changing job conditions, and lack of development was significantly associated with intention to leave the current job by controlling for confounding variables in the regression model (Table 4).

## Discussion

The health workforce is the most imperative building block of the health system [26], without which the other five building blocks, such as service delivery, health information system leadership and governance, and access to essential medicine, cannot realize quality public health services [27, 28]. In relation to this, widespread nursing shortages and nurses' high turnover rates have become global issues [4]. To bring the solution for this devastating issue, identifying the magnitude of the problem is the core. As part of future evidence and solutions, this study intends to assess nurses' job satisfaction and intention to leave their current job and the associated factors in federal public hospitals in Addis Ababa, Ethiopia.

Job satisfaction is an essential factor that affects employees' initiative and enthusiasm [29]. The relationship between job satisfaction and employee intention to leave their job is well

**Table 3. Work environment characteristics among nurses working in selected federal hospitals in Ethiopia, 2020 (N = 392).**

| Variables | Category | Frequency | Percent |
|---|---|---|---|
| Staffing is correct in ratio to the workload experiencing | Yes | 209 | 53.3 |
|  | No | 183 | 46.7 |
| Feel that nursing education matches the practical work | Yes | 324 | 82.7 |
|  | No | 68 | 17.3 |
| Have support from coworkers | Yes | 326 | 83.2 |
|  | No | 66 | 16.8 |
| Find the job stressful | Yes | 237 | 60.5 |
|  | No | 155 | 39.5 |
| Have thoughts about leaving an organization | Yes | 267 | 68.1 |
|  | No | 125 | 31.9 |
| Have thoughts about leaving the profession | Yes | 198 | 50.5 |
|  | No | 194 | 49.5 |
| Applied for a new job during the last year | Yes | 200 | 51.0 |
|  | No | 192 | 49.0 |
| Salary is the reason thinking about changing the job | Yes | 283 | 72.2 |
|  | No | 109 | 27.8 |
| Schedule/working hours are reason why have thoughts about quitting the current job | Yes | 256 | .3 |
|  | No | 136 | 34.765 |
| Manager is the reason for thoughts about leaving job | Yes | 201 | 51.3 |
|  | No | 191 | 48.7 |
| Organizational factors are reason for having thoughts about quitting the job | Yes | 301 | 76.8 |
|  | No | 91 | 23.2 |
| Career is a factor for thinking thoughts about changing the job | Yes | 241 | 61.5 |
|  | No | 151 | 38.5 |
| Health is a reason for having thoughts about changing the job | Yes | 203 | 51.8 |
|  | No | 189 | 48.2 |
| Lack of development is a reason for thinking thoughts about changing the job | Yes | 146 | 62.8 |
|  | No | 246 | 37.2 |

explained in different studies [7, 30, 31]. This literature shows that one of the major reasons for workers to leave their job is dissatisfaction with their current job for different reasons [1, 6]. According to the results of this study, the magnitude of satisfaction of their current job among nurses in the selected hospitals was only 47.6%. This finding is almost consistent with a pooled prevalence of job satisfaction among health professionals in Ethiopia [12]. Conversely, the result is much lower than the result of a study performed in a tertiary center in Keno, Northwest Nigeria, in which the magnitude of satisfaction of nurses with their jobs was 90.4% [30]. The result is also lower than the result of a study performed in Turkey among nurses [31]. The difference might be due to variation in the data collection tools, use of different cut-off points, sample size, sampling methods, study setting and study participant variation.

Different studies show that the intention to leave a job is the most important and immediate antecedent of turnover decisions [1, 28]. When health workers leave their current jobs, it can negatively affect organizational performance and contribute to the shortage of the health work force [1, 11, 19]. According to the results of this study, approximately 80.6% of the study population intended to leave their current job. This result is almost consistent with the findings of a study performed in a health institution in Addis Ababa, Ethiopia [20]. However, it is much higher than the finding of a study performed in a public health center in the Jimma Zone of southwestern Ethiopia, which found 63.7% intention to leave their current job among health

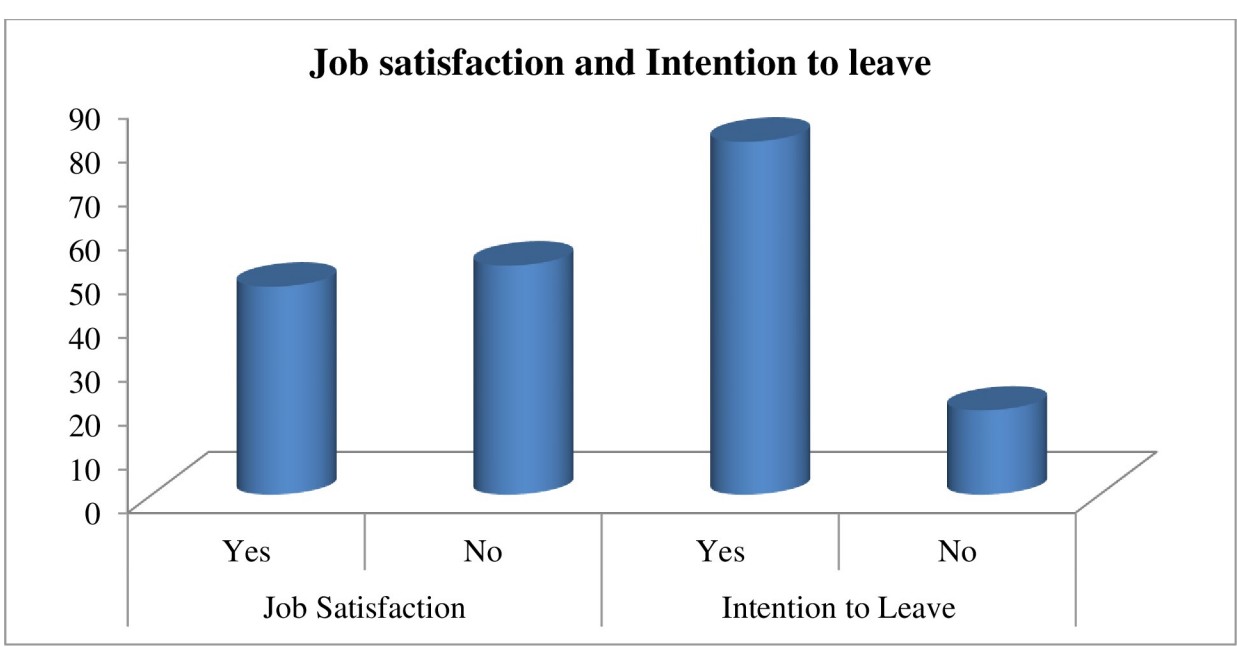

**Fig 1. Job satisfaction and intention to leave among nurses working in selected federal hospitals in Ethiopia, 2020 (N = 392).**

care professionals. This finding is also higher than the results of studies performed in China [4] and Addis Ababa [19].

When talent employees leave their jobs, the organizations will lose their experience and may face a problem with the additional costs spent recruiting, training and developing new candidates [32, 33]. To retain experienced employees, organizational managers need to have a strong plan for developing skills/competence/qualification for their own and employees [33]. As such, this study also shows a strong relationship between the availability of manager plans to develop their competence and intention to leave. The intention to leave the current job of those who responded that the manager had no personal plan to develop his/her skill/competence/qualification was 3.74 times higher.

Different studies have identified a direct relationship between job stress and intention to leave [32, 34]. Stress is a component of every human existence, and a human faces a sequence of stress in his everyday life. It becomes a threat when a person faces it at work; thus, it becomes an essential concern for the employer and the employees as well [35]. The results of this study also support this reality by indicating a strong relationship between the two. Those respondents who said the job was stressful had a 28% higher rate of intention to leave their current job. This suggests that the organization should identify the major causes of stress at the workplace and plan and implement the possible mitigation systems.

Salary is one of the basic issues in organizations for employees to work hard and stay in the organizations for a long period of time [2, 36]. Employees will be much more invested in their jobs and in the company if they feel valued by that organization. A higher salary is a way to show employees that they are valued. Companies can also demand higher quality work and higher levels of productivity and services in exchange for that higher salary. According to this study, salary is among the significant determining factors for employees to leave or stay at their current job. Accordingly, those who said salary is not in balance with the demands of employees had a 2.85 times higher rate of intention to leave their current jobs. This study is supported by many studies [19, 21, 26]. Therefore, this confirms that salary is one of the best ways to compensate for employees' time and efforts invested in the organization.

**Table 4. Factors associated with intention to leave their current job among nurses working in selected federal hospitals in Ethiopia, 2020 (N = 392).**

| Variables | Responses | Intention to leave | | COR (95%CI) | AOR (95%CI) |
|---|---|---|---|---|---|
| | | Yes N (%) | No N (%) | | |
| Satisfied with the amount of compensation paid | Yes | 120 (75.9) | 38 (24.1) | 0.61 (0.37, 1.01) | 0.58 (0.25, 1.37) |
| | No | 196 (83.8) | 38 (16.2) | 1.00 | 1.00 |
| Feeling of wrong career | Yes | 141 (84.9) | 25 (15.1) | 1.64 (0.97, 2.79) | 0.55 (0.20, 1.47) |
| | No | 175 (77.4) | 51 (22.6) | 1.00 | 1.00 |
| Salary is imbalance to the demands | Yes | 155 (89.6) | 18 (10.4) | 3.10 (1.75, 5.50) | 2.85 (1.24, 6.57)* |
| | No | 161 (73.5) | 58 (26.5) | 1.00 | 1.00 |
| Experience lack of support from management | Yes | 219 (83.0) | 45 (17.0) | 1.56 (0.93, 2.61) | 2.00 (0.87, 4.62) |
| | No | 97 (75.8) | 31 (24.2) | 1.00 | 1.00 |
| Clear communication from the manager | Yes | 145 (86.3) | 23 (13.7) | 1.95 (1.14, 3.34) | 1.10 (0.50, 2.41) |
| | No | 171 (76.3) | 53 (23.7) | 1.00 | 1.00 |
| Manager has a personal plan for developing skill/competence/ | Yes | 168 (90.8) | 17 (9.2) | 1.00 | 1.00 |
| | No | 148 (71.5) | 59 (28.5) | 3.94 (2.20, 7.06) | 3.74 (1.58, 8.87)* |
| Support from co-workers | Yes | 254 (77.9) | 72 (22.1) | 0.23 (0.08, 0.65) | 0.31 (0.09, 1.08) |
| | No | 62 (93.9) | 4 (6.1) | 1.00 | 1.00 |
| Stressful job | Yes | 186 (78.5) | 51 (21.5) | 1.00 | 1.00 |
| | No | 130 (83.9) | 25 (16.1) | 0.70 (0.41, 1.19) | 0.28 (0.11, 0.71)* |
| Schedule/working hours is a reason for changing the job | Yes | 229 (89.5) | 27 (10.5) | 4.77 (2.81, 8.12) | 0.72 (0.29, 1.79) |
| | No | 87 (64.0) | 49 (36.0) | 1.00 | 1.00 |
| Manager is the reason for thoughts about leaving the job | Yes | 185 (92.0) | 16 (8.0) | 5.30 (2.92, 9.60) | 1.33 (0.55, 3.21) |
| | No | 131 (68.6) | 60 (31.4) | 1.00 | 1.00 |
| Organizational factors are reason for having thoughts about leaving the job | Yes | 271 (90.0) | 30 (10.0) | 9.23 (5.29, 16.1) | 1.01 (0.36, 2.83) |
| | No | 45 (49.5) | 46 (50.5) | 1.00 | 1.00 |
| Career is a factor for thoughts about changing the job | Yes | 218 (90.5) | 23 (9.5) | 5.13 (2.97, 8.84) | 0.72 (0.28, 1.91) |
| | No | 98 (64.9) | 53 (35.1) | 1.00 | 1.00 |
| Health problem is a reason for having thoughts about changing the job | Yes | 185 (91.1) | 18 (8.9) | 4.55 (2.56, 8.08) | 3.02 (1.17, 7.79)* |
| | No | 131 (69.3) | 58 (30.7) | 1.00 | 1.00 |
| Lack of development is a reason for thoughts about changing the job | Yes | 225 (91.5) | 21 (8.5) | 6.48 (3.70, 11.3) | 4.13 (1.51, 11.3)* |
| | No | 91 (62.3) | 55 (37.7) | 1.00 | 1.00 |

*P value<0.05

**p value <0.01, CI: confidence interval, COR: crude odds ratio, AOR: adjusted odds ratio

Different evidence shows that, with established and effective training and education programs, an institution can improve retention and increase staff morale by creating a positive, motivated, and competent workforce [37, 38]. This, in turn, improves patient satisfaction and profitability of the organization [2, 6]. Similar to this evidence, the results of this study also show that there is a strong relationship between intentions to leave one's current job and a lack of personal development opportunities. This result is supported by numerous research findings [12, 20, 35].

## Conclusion and recommendation

The overall intention to leave a current job among nurses working in selected federal hospitals was high. The salary being imbalanced with the demands, managers having a personal plan for developing skills/competence/qualifications, stressfulness of the jobs, health problems and lack of development were the determinant factors for nurses' intention to leave their current job.

The government of Ethiopia should balance salaries of nurses with the current market level. Hospitals leaders should plan the way nurses develop their educational and job carrier levels. Effective measures should be taken to improve hospital nurses' accomplishment, professional status, and career development to minimize their current intention to leave. In general, the organization should plan the way to retain their experienced workers for a long period of time to reach the planned vision. Future researchers should conduct longitudinal and interventional studies to assess the definite intention to leave.

## Supporting information

**S1 Data.**
(SAV)

## Acknowledgments

Our special thanks go to St. Paul's Hospital and St. Peter's hospital was included in the study for their willingness to give us supportive letters and important information for our work. Lastly, we would like to thank all participants included in the study for their willingness to participate.

## Author Contributions

**Conceptualization:** Aynye Negesse Woldekiros, Elsabet Getye, Ziyad Ahmed Abdo.

**Data curation:** Aynye Negesse Woldekiros, Elsabet Getye, Ziyad Ahmed Abdo.

**Formal analysis:** Aynye Negesse Woldekiros, Elsabet Getye, Ziyad Ahmed Abdo.

**Funding acquisition:** Elsabet Getye.

**Investigation:** Aynye Negesse Woldekiros, Elsabet Getye, Ziyad Ahmed Abdo.

**Methodology:** Aynye Negesse Woldekiros, Elsabet Getye, Ziyad Ahmed Abdo.

**Project administration:** Aynye Negesse Woldekiros, Elsabet Getye.

**Resources:** Elsabet Getye, Ziyad Ahmed Abdo.

**Software:** Aynye Negesse Woldekiros, Ziyad Ahmed Abdo.

**Supervision:** Aynye Negesse Woldekiros, Ziyad Ahmed Abdo.

**Validation:** Aynye Negesse Woldekiros, Elsabet Getye, Ziyad Ahmed Abdo.

**Visualization:** Elsabet Getye.

**Writing – original draft:** Aynye Negesse Woldekiros, Elsabet Getye, Ziyad Ahmed Abdo.

**Writing – review & editing:** Aynye Negesse Woldekiros, Elsabet Getye, Ziyad Ahmed Abdo.

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
