## [Decision Letter · Decision Letter 0]

24 Jan 2022

PONE-D-21-26236

MAGNITUDE OF JOB SATISFACTION AND INTENTION TO LEAVE PRESENT JOB AMONG NURSES IN SELECTED FEDERAL HOSPITALS IN ADDIS ABABA, ETHIOPIA

PLOS ONE

Dear Dr. Woldekiros,

Thank you for submitting your manuscript to PLOS ONE. After careful consideration, we feel that it has merit but does not fully meet PLOS ONE’s publication criteria as it currently stands. Therefore, we invite you to submit a revised version of the manuscript that addresses the points raised during the review process.

The revised version should address all comments.

There is a substantial body of literature on quit intentions that is not covered in the manuscript (see https://doi.org/10.1111/j.1468-232X.2008.00546.x). This should be stated in the paper. 

We look forward to receiving your revised manuscript.

Kind regards,

Petri Böckerman

Academic Editor

PLOS ONE

Journal Requirements:

7. Your ethics statement should only appear in the Methods section of your manuscript. If your ethics statement is written in any section besides the Methods, please move it to the Methods section and delete it from any other section. Please ensure that your ethics statement is included in your manuscript, as the ethics statement entered into the online submission form will not be published alongside your manuscript. 

8. We note you have included a table to which you do not refer in the text of your manuscript. Please ensure that you refer to Table 3 in your text; if accepted, production will need this reference to link the reader to the Table.

9. We noticed you have some minor occurrence of overlapping text with the following previous publication(s), which needs to be addressed:

- https://www.science.gov/topicpages/s/satisfaction+turnover+intentions.html

- https://cyberleninka.org/article/n/4194

- https://www.hindawi.com/journals/bmri/2019/7092964/

The text that needs to be addressed involves the Introduction.

In your revision ensure you cite all your sources (including your own works), and quote or rephrase any duplicated text outside the methods section. Further consideration is dependent on these concerns being addressed.

Reviewers' comments:

Reviewer's Responses to Questions

**Comments to the Author**

1. Is the manuscript technically sound, and do the data support the conclusions?

Reviewer #1: Partly

Reviewer #2: Yes

2. Has the statistical analysis been performed appropriately and rigorously? 

Reviewer #1: Yes

Reviewer #2: Yes

3. Have the authors made all data underlying the findings in their manuscript fully available?

Reviewer #1: Yes

Reviewer #2: Yes

4. Is the manuscript presented in an intelligible fashion and written in standard English?

Reviewer #1: Yes

Reviewer #2: No

5. Review Comments to the Author

Reviewer #1: There is substantial literature published on the area present by the manuscript. Examples of recent published literature on this area of research include:

Worku N, Feleke A, Debie A, Nigusie A. Magnitude of Intention to Leave and Associated Factors among Health Workers Working at Primary Hospitals of North Gondar Zone, Northwest Ethiopia: Mixed Methods. BioMed research international. 2019 Jul 16;2019.

Worku N, Feleke A, Debie A, Nigusie A. Magnitude of Intention to Leave and Associated Factors among Health Workers Working at Primary Hospitals of North Gondar Zone, Northwest Ethiopia: Mixed Methods. BioMed research international. 2019 Jul 16;2019.

Dado WM, Mekonnen W, Aragw MD, Desta BF, Desal AY. Turnover intention of health workers in public-private mix partnership health facilities: A case of Addis Ababa, Ethiopia. Epidemiology (Sunnyvale). 2019 Jun 7;9(2):374.

There is very little that the findings of the study at to the existing database. No cause-effect relation can be established due to study design limitations.

Reviewer #2: This is an interesting paper depicting the loss of health care workers and the reasons for mass resignation. It is a relevant topic which must be brought to the attention of management and like minded individuals. While the technicalities of the paper were simple, the underlying message is clear. For the paper to be accepted, a native English speaker should review the paper to improve the language. I also highly recommend a thorough formatting of the paper for publication in the journal. There is also a missed opportunity to discuss further the sort of interventions that might allow for prevention of loss of nurses.

6. PLOS authors have the option to publish the peer review history of their article (what does this mean?). If published, this will include your full peer review and any attached files.

Reviewer #1: No

Reviewer #2: No

---

## [Author Response · Author response to Decision Letter 0]

5 Mar 2022

1. Comments from reviewer 1:

Comment 1: There is substantial literature published on the area present by the manuscript. 

Response: Thank you for your valuable comment. We agree that there are some literatures done in this area previously. However, the literatures published are not enough to show this sound problem in the country, specifically study area. Addis Ababa is a place where huge and complex health problems are screened and treated. This needs an adequate number of well experienced health care professionals to appropriately respond those public health problems. As such, hospitals need to retain experienced health care professionals including nurses. To make this practical, additional evidences are needed to know the overall prevalence of the problem (Pooled prevalence), and suggest to policy makers for interventions. 

Comment 2: There is very little that the findings of the study at to the existing database. 

Response: Thank you again for the comment: We disagree with the comment. We belief that the finding of this study will provided additional input to the existing literature to show the overall magnitude of the problem. 

Comment 3: No cause-effect relation can be established due to study design limitations

Response: Thank you again for additional input: We agreed with the comment. Due to design limitation, the study do not established cause-effect relationship; however, we recommended longitudinal and interventional study for future researches to fill this drawback. 

2. Comment from reviewer 2: 

Comment 1: This is an interesting paper depicting the loss of health care workers and the reasons for mass resignation. It is a relevant topic which must be brought to the attention of management and likeminded individuals. While the technicalities of the paper were simple, the underlying message is clear. 

Response: Thank you for the interesting understanding and explanation of the problem. 

Comment 2: For the paper to be accepted, a native English speaker should review the paper to improve the language.

Response: We have used experienced professionals to edit the paper from both English language and content point of view. 

Comment 3: I also highly recommend a thorough formatting of the paper for publication in the journal. 

Response: We have tried to format the paper according to the journal formatting style. 

Comment 4: There is also a missed opportunity to discuss further the sort of interventions that might allow for prevention of loss of nurses.

Response: Thank you for interesting point. We have tried to discuss possible sort of interventions in both discussion and recommendation parts.

---

## [Decision Letter · Decision Letter 1]

28 Mar 2022

PONE-D-21-26236R1Magnitude of job satisfaction and intention to leave present job among nurses in selected Federal Hospitals in Addis Ababa, EthiopiaPLOS ONE

Dear Dr. Woldekiros,

Thank you for submitting your manuscript to PLOS ONE. After careful consideration, we feel that it has merit but does not fully meet PLOS ONE’s publication criteria as it currently stands. Therefore, we invite you to submit a revised version of the manuscript that addresses the points raised during the review process. The revised version should address the remaining concerns. Please submit your revised manuscript by May 12 2022 11:59PM. If you will need more time than this to complete your revisions, please reply to this message or contact the journal office at plosone@plos.org. Please include the following items when submitting your revised manuscript:A rebuttal letter that responds to each point raised by the academic editor and reviewer(s). You should upload this letter as a separate file labeled 'Response to Reviewers'.A marked-up copy of your manuscript that highlights changes made to the original version. You should upload this as a separate file labeled 'Revised Manuscript with Track Changes'.An unmarked version of your revised paper without tracked changes. You should upload this as a separate file labeled 'Manuscript'.If applicable, we recommend that you deposit your laboratory protocols in protocols.io to enhance the reproducibility of your results. Protocols.io assigns your protocol its own identifier (DOI) so that it can be cited independently in the future. For instructions see: https://journals.plos.org/plosone/s/submission-guidelines#loc-laboratory-protocols. Additionally, PLOS ONE offers an option for publishing peer-reviewed Lab Protocol articles, which describe protocols hosted on protocols.io. Read more information on sharing protocols at https://plos.org/protocols?utm_medium=editorial-email&utm_source=authorletters&utm_campaign=protocols.

We look forward to receiving your revised manuscript.

Kind regards,

Petri Böckerman

Academic Editor

PLOS ONE

Journal Requirements:

Reviewers' comments:

Reviewer's Responses to Questions

**Comments to the Author**

1. If the authors have adequately addressed your comments raised in a previous round of review and you feel that this manuscript is now acceptable for publication, you may indicate that here to bypass the “Comments to the Author” section, enter your conflict of interest statement in the “Confidential to Editor” section, and submit your "Accept" recommendation.

Reviewer #2: (No Response)

2. Is the manuscript technically sound, and do the data support the conclusions?

Reviewer #2: Yes

3. Has the statistical analysis been performed appropriately and rigorously? 

Reviewer #2: Yes

4. Have the authors made all data underlying the findings in their manuscript fully available?

Reviewer #2: Yes

5. Is the manuscript presented in an intelligible fashion and written in standard English?

Reviewer #2: No

6. Review Comments to the Author

Reviewer #2: While the authors have tried to address most of the queries from the editor and the reviewers, the quality of language is still poor in the manuscript. Authors are advised to check on acronyms, capitalization, tense and sentence structure to be published in a international journal. Again I advise a professional English service for this.

7. PLOS authors have the option to publish the peer review history of their article (what does this mean?). If published, this will include your full peer review and any attached files.

Reviewer #2: No

---

## [Author Response · Author response to Decision Letter 1]

9 May 2022

Comment from reviewer #2

• While the authors have tried to address most of the queries from the editor and the reviewers, the quality of language is still poor in the manuscript. Authors are advised to check on acronyms, capitalization, tense and sentence structure to be published in a international journal. Again I advise a professional English service for this.

Response to reviewer #2

• As the comments of reviewer #2, we tried to improve the quality of the manuscript’s language. We check all the acronyms, capitalization, tense and sentence structure as the quality of publication permits. Furthermore, we tried to get professional English language support from American Journal Experts.

---

## [Decision Letter · Decision Letter 2]

24 May 2022

Magnitude of job satisfaction and intention to leave their present job among nurses in selected Federal Hospitals in Addis Ababa, Ethiopia

PONE-D-21-26236R2

Dear Dr. Woldekiros,

We’re pleased to inform you that your manuscript has been judged scientifically suitable for publication and will be formally accepted for publication once it meets all outstanding technical requirements.

Kind regards,

Petri Böckerman

Academic Editor

PLOS ONE

Additional Editor Comments (optional):

Reviewers' comments:

Reviewer's Responses to Questions

**Comments to the Author**

1. If the authors have adequately addressed your comments raised in a previous round of review and you feel that this manuscript is now acceptable for publication, you may indicate that here to bypass the “Comments to the Author” section, enter your conflict of interest statement in the “Confidential to Editor” section, and submit your "Accept" recommendation.

Reviewer #2: All comments have been addressed

2. Is the manuscript technically sound, and do the data support the conclusions?

Reviewer #2: (No Response)

3. Has the statistical analysis been performed appropriately and rigorously? 

Reviewer #2: (No Response)

4. Have the authors made all data underlying the findings in their manuscript fully available?

Reviewer #2: (No Response)

5. Is the manuscript presented in an intelligible fashion and written in standard English?

Reviewer #2: (No Response)

6. Review Comments to the Author

Reviewer #2: (No Response)

7. PLOS authors have the option to publish the peer review history of their article (what does this mean?). If published, this will include your full peer review and any attached files.

Reviewer #2: No

---

## [Editor Report · Acceptance letter]

30 May 2022

PONE-D-21-26236R2 

Magnitude of job satisfaction and intention to leave their present job among nurses in selected Federal Hospitals in Addis Ababa, Ethiopia 

Dear Dr. Woldekiros:

I'm pleased to inform you that your manuscript has been deemed suitable for publication in PLOS ONE. Congratulations! Your manuscript is now with our production department. 

Kind regards, 

on behalf of

Professor Petri Böckerman 

Academic Editor

PLOS ONE